# Optimal Calibration of Evaporation Models against Penman–Monteith Equation

**Dagmar Dlouhá** [ID], **Viktor Dubovský** *[ID] **and Lukáš Pospíšil** [ID]

Department of Mathematics, Faculty of Civil Engineering, VSB-TU Ostrava, Ludvíka Podéště 1875/17, 708 00 Ostrava, Czech Republic; dagmar.dlouha@vsb.cz (D.D.); lukas.pospisil@vsb.cz (L.P.)
* Correspondence: viktor.dubovsky@vsb.cz

**Abstract:** We present an approach for the calibration of simplified evaporation model parameters based on the optimization of parameters against the most complex model for evaporation estimation, i.e., the Penman–Monteith equation. This model computes the evaporation from several input quantities, such as air temperature, wind speed, heat storage, net radiation etc. However, sometimes all these values are not available, therefore we must use simplified models. Our interest in free water surface evaporation is given by the need for ongoing hydric reclamation of the former Ležáky–Most quarry, i.e., the ongoing restoration of the land that has been mined to a natural and economically usable state. For emerging pit lakes, the prediction of evaporation and the level of water plays a crucial role. We examine the methodology on several popular models and standard statistical measures. The presented approach can be applied in a general model calibration process subject to any theoretical or measured evaporation.

**Keywords:** model calibration; evaporation; Penman–Monteith equation; optimization; cross-validation

## 1. Introduction

Evaporation and evapotranspiration play a crucial role in water management in a wide range of human activities and thus there is a strong need for accurate estimates. This need leads to a considerable number of papers and studies that are offering new methods of such estimates or comparison of methods already used in hydrologic engineering applications. The results of estimation could be compared with reference evaporation/evapotranspiration calculated by FAO Penman–Monteith equation $E_{FAO}$, which is recommended as the standard method [1,2]. This equation is considered to be an *etalon* to which the results of the other methods can be related and compared. The papers dealing with evaporation or evapotranspiration present the comparison of the *FAO Penman–Monteith* method results to other methods proposing the relations between input data less complicated and computationally less demanding. For instance, such a procedure was performed in the study [3] to find the best estimation of water lost from a covered reservoir. In [4] it is stated that $E_{FAO}$ provides good agreement with evaporation measured on 120 ha dam. In [5], $E_{FAO}$ is used not only as a reference method and but also a foundation of new numerical models derived by multiple linear regression and design of experiment method, followed by the simplified methodology for the quantification of the evaporation rate of a basin with a photovoltaic system. Similarly, the possibility of reduction of Lake Nasser evaporation using a floating photovoltaic system is described in [6]. Our research is not focused only on the area of Lake Most, but also on pit lakes that are only planned, therefore it is impossible to use limnological and bathymetric data, such as temperature profile or water depth. Such a situation is considered in [7], where authors used the $E_{FAO}$ for the computation of the open water evaporation estimation.

FAO Penman–Monteith method is characterized by a strong likelihood of correctly predicting evapotranspiration in a wide range of locations and climates with differing

local conditions, e.g., solar radiation, sunshine duration, wind speed, air humidity, air temperature [8–10].

Our interest in free water surface evaporation is given by the need for ongoing hydric recultivation of the former Ležáky–Most quarry (Czech Republic), i.e., Lake Most, as well as another planned hydric recultivation in the region. One of the key components of hydric reclamation planning is the securitization of long-term sustainability, which is based on the capability of keeping the stable level of a dimension of the final water level.

Hydric recultivation was proposed to be the best way to deal with the residual of open-cut coal mines in the north-western region of the Czech Republic. After the mine is closed, the void could be filled by surface water runoff and groundwater. In the case that these resources are not strong enough, the pit lake has to be filled artificially and that is the case of the former Most-Ležáky mine and Lake Most. The level of the new lake has been proposed to be stable with a water level at 199 m above sea level assuming the tabulated values of precipitation and evaporation between 500 mm and 600 mm annually [11]. However, this assumption fails to be true [12].

The evolution of artificial pit lakes is affected by a wide range of chemical, physical, and namely hydrological processes such as saturation of the coastal lines, leakage through the bottom, and free surface evaporation. As the lake bottom was sealed before filling, evaporation was supposed to be the main cause of the observed water loss.

Together with the precipitation, the open water *evaporation* and vegetation *evapotranspiration* form the main components of the water cycle in nature, and it is said that the evaporation over the land surface amounts to about two thirds of the average precipitation, see [13]. However, these estimates differ from location to location and evaporation measurement or calculation procedures are complicated and burdened with a high degree of uncertainty. This happens due to the complexity of evaporation as a physical phenomenon and several factors that affect this process. The rate of evaporation could be measured or calculated, however, because of the simplification of the evaporation process description, both measurement and computational methods provide only the approximation of actual evaporation. For further discussion about the historical development of *evaporation* and *evapotranspiration*, see [14] mentioning 166 models and equations obtained during the last three centuries. The description and characterization of all physical processes affecting evaporation could be found in a classical book by Brutsaert [15], or Maidment [16] ( Shuttleworth's chapter).

For computational methods, there are two ways to handle evaporation, described as *mass transfer* or *energy budget* methods, see [13]. Furthermore, the models could be viewed as *temperature-based*, *radiation-based*, *mass transfer-based*, and *combined methods* based on the inputs used to calculate the rate of evaporation, for additional details see [2,16–18].

Additionally, within each group, there are several equations, which are widely cited in the technical literature. During the study of various evaporation models in the literature, one can observe several difficulties. One of the main difficulties is the inconsistencies in the used physical units. For instance, according to the time and place of publication of articles or books, the same equation can be encountered with the pressure given in kPa, mbar, Torr or *millimeters of mercury column*. This variability of units can cause different shapes of the same equations in different sources, even when the same units are used. Furthermore, different types of methods require different type of data. However, in practical applications, we are not able to measure all types of input parameters, therefore the choice of the model depends not only on the modelling quality but mainly on the ability to measure the required input physical quantities.

This is one of the main reasons why the development of new simplified models is still active. The FAO equation is a solid standard, but sometimes too complex to be handled in practice. For instance, in the article [19], eighteen *temperature-*, *radiation-*, *mass transfer*-based and *combined methods* are studied under the condition of climatic change in Germany. The FAO equation is compared with 31 methods under the *humid* climate condition in Iran

in the paper [20]. In [21], five temperature-based methods and three radiation methods are considered.

In this paper, we are looking for the simplification of the FAO equation in terms of the number of input quantities. Our goal is to use less complex models to model the evaporation in the area of Lake Most by calibrating the parameters of the models in the fitting optimization process against the evaporation estimation by FAO using selected statistical measures. The motivation came from the lack of measuring devices in the direct area of the lake. In this paper, we consider models which require only the air temperature, wind speed, and relative humidity. Besides the Lake Most, we are interested also in the planned pit lakes. Therefore, the types of models are limited to those which requires only these basic meteorological data. In this case, it is not possible to measure, for instance, the temperature of the water.

The performance of the considered methods can be evaluated statistically, and several statistical measures could be used. The most commonly used are the Root Mean Square Error (RMSE), Mean absolute error (MAE), and the Mean Bias Error (MBE), see, for instance, [22]. Additionally, the RMSE and MBE could be combined to calculate the so-called *t*-statistic test, which expresses the level of confidence between the models, see [23]. A quite unusual measure could be found in [24], it is *mean ratio MR*, which is computed as the average of ratios between the predicted and observed values. Another way to compare two given models is to compute *Pearson's* correlation coefficient (PCC) *r* or to compute $R^2$ coefficient of determination. These two measures are closely connected since $R^2$ is square of *r*. The simplest measure of the prediction quality is the percentage expression of the difference between models. This is a statistical measure that is well known as percent bias (PBIAS). The last class of statistical measures that describe the correspondence of observed and simulated data are agreement indices, such as *Nash–Sutcliffe* efficiency (NSE) and *Willmot's* agreement index. These are used, for example, in [21,25,26]. For further discussion of statistical model evaluations, see [27]. In this paper, we compare several selected statistical measures, namely $NSE, RMSE, MAE$, and $PBIAS$. However, we suppose that our methodology can be applied to any chosen distance function. The applicability depends on the ability to solve the corresponding regression problem.

The choice of temperature-based methods in the case of the Lake Most study is not due to the *current* lack of meteorological data since *Kopisty* meteostation is located only 1 km from the lake. The simplification of the equation is motivated by the further planned *hydric recultivations* in the region. Planned pit lakes are more distant from *Kopisty* and therefore the meteorological data provided to the models from *Kopisty* would not be sufficient. The data provided to the models on new lakes will be measured directly on the area of new lakes. In the case of Lake Most, we can identify the appropriate simplified model because of advantageous location of *Kopisty* with respect to Lake Most. Therefore, we are interested in the identification of the simplest suitable model with the low demand on input data. The construction of the new weather station in the area closer to the new lake does not make any sense from the financial point of view (because of the presence of *Kopisty* weather station). On the other hand, to provide better estimations, we should measure the input data as close as possible to the area of interest. The measurement of, for instance, the temperature is relatively cheap. The only question is if the temperature is a sufficient amount of input meteorological data for providing a sufficient estimation. The accuracy of the water loss due to evaporation is crucial for those planned artificial lakes. To provide the best possible estimate, the experiences from the Lake Most will be used. In this paper, we compare several simplified models with respect to different statistical measures, namely $NSE, RMSE, MAE$, and $PBIAS$.

Additionally, to avoid the overfitting of the calibrated model, we adopt the cross-validation methodology [28]. We randomly split the data into calibration and validation parts. The parameters of the model are optimized on the calibration set and tested on the validation part. Results from the validation part are further analyzed and the best model is chosen concerning results from all cross-validation splittings.

The paper is organized as follows. Section 2.4 introduces the methods and materials used in our computation. To be more specific, we start with the presentation of the Lake Most in Section 2.1, and the data provided to the models in Section 2.2. Afterwards, we review the FAO equation in Section 2.3 and the simplified models in Section 2.4. During the calibration process, we use the statistical measures presented in Section 2.5. The whole methodology is implemented in R programming language, see Section 2.6 for details. This section also includes the description of the used cross-validation process. The results are presented in Section 3 and discussed in Section 4. Finally, Section 5 concludes the paper.

The paper can be considered to be an extension of our previously published work [29].

## 2. Materials and Methods

### 2.1. Study Area

The Lake Most is situated in the North of the Czech Republic near the city of Most 50°C310′ N, 13°C360′ E, see Figure 1. It was created by the hydric recultivation of the *Most–Ležáky* quarry in the central part of the *North Bohemian* brown coal basin. The former mine heavily affected the area of 1254 ha and the pit lake, as a part of its revitalization, was planned to have a surface area of about 300 ha. The project of the revitalization is secured by the state enterprise *Palivový kombinát Ústí (PKU)* [30].

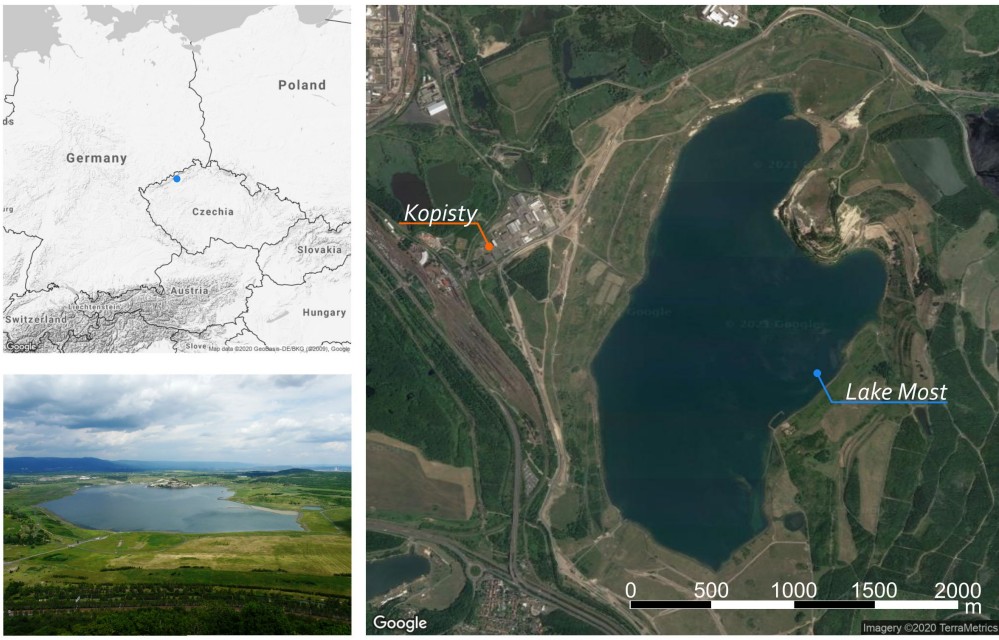

**Figure 1.** Lake Most and the surrounding area (source: www.pku.cz, Google Earth).

Before the flooding, it was necessary to take technical arrangements such as sealing the bottom of the future lake, construction of an underground sealing wall, and strengthening the shoreline. All these arrangements allow viewing the Lake Most as a closed system without natural inflow or outflow. Due to the absent natural inflow, the residual pit of the lake was filled through an artificial feeder during the period from 2008 to 2014. In the final phase of lake filling, i.e., in the year 2014, the surface level rose from 197.74 m to the required level of 199 m above sea level.

After finishing the filling process, Lake Most has an actual surface area of 309.4 ha, a coastal line length of 8.9 km, a total water volume of 70.5 million m$^3$, and a maximum depth of 75 m. Throughout the filling of the lake, both *operational* and *basic meteorological* data were monitored. The operational data contain data on the achieved altitude of the lake level, its surface area, and especially on the volume of water admitted. The filling of the lake has been finished in 2014 achieving the required surface level of 199 m.

### 2.2. Data and Data Sources

In our research, we are using the meteorological data collected during the years 2015–2019. The collection includes all data necessary for the calculation of the Penman–Monteith equation (see Section 2.3). These meteorological measurements were performed at the *Kopisty* weather station situated approximately 1 km from the lake. The station is operated by *CHMI—Czech Hydrometeorological Institute* and the data are recorded at ten-minute intervals. The dataset obtained from *CHMI* was statistically processed to be used in the equations to model the evaporation. We present the data basic statistics in Figure 2. In *Kopisty* weather station, the wind speed is measured at 10 m above the ground to avoid the influence of the ground. The air temperature and humidity are measured at 2 m above the ground.

We also included precipitation frequency for the demonstration of the hydrological balance in the area of interest. In comparison with the average temperature and precipitation in the Czech Republic, the area of the planned hydric reclamation is in the area with the temperature strongly above the average and precipitation strongly below normal precipitation, and with the number of hours of sunshine below the typical value in the Czech Republic [11].

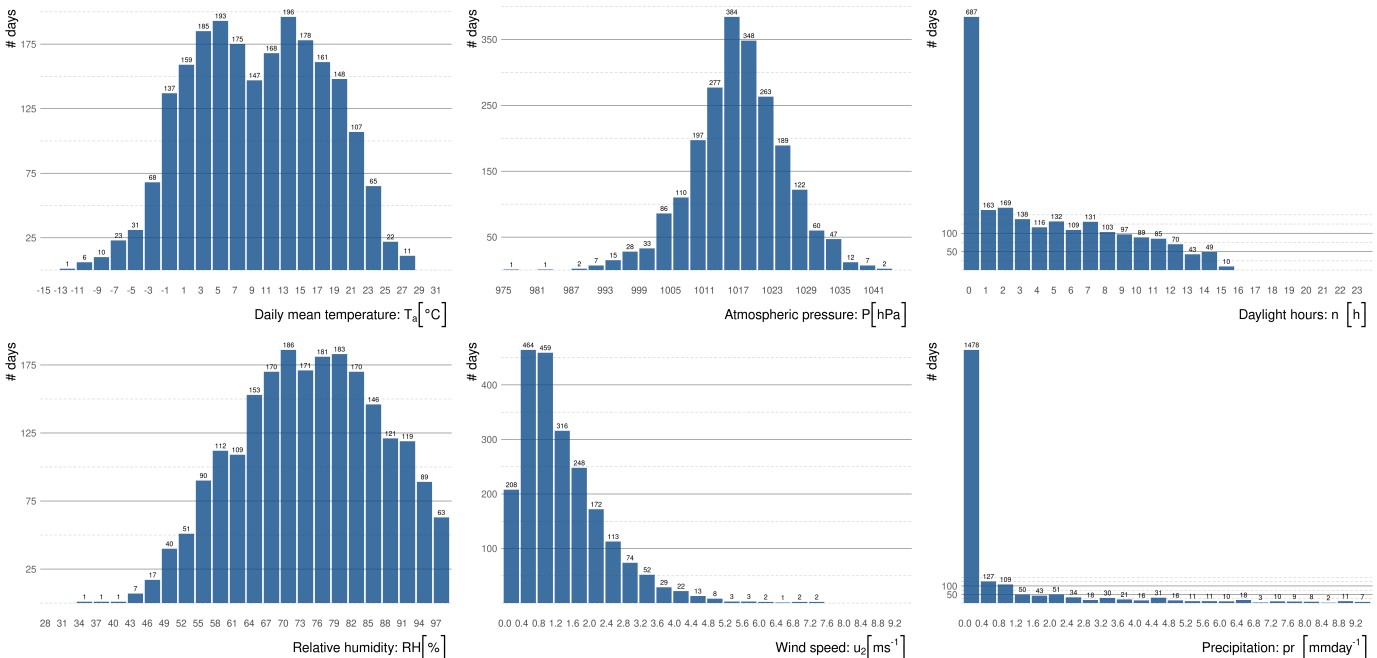

**Figure 2.** The basic statistics of the daily measurements performed at the *Kopisty* weather station during the years 2015–2019: average day temperature $T_a$, atmospheric pressure $P$, daylight hours per day $n$, relative humidity $RH$, wind speed $u_2$, and precipitation $pr$. These data are used in the equations for modelling the evaporation.

### 2.3. Penman–Monteith Equation

The $E_{FAO}$ equation is of the form

$$E_{FAO} = \frac{0.408\,\Delta\,(R_n - G) + \gamma\,\frac{900}{T_a+273}\,u_2(e_s - e_a)}{\Delta + \gamma\,(1 + 0.34\,u_2)}. \tag{1}$$

Please see Section Abbreviations at the end of this paper for the description and physical units of the used variables.

According to *Linacre* paper [31] for daily estimates of the evaporation rate of free water level, the term $G$ can be neglected, i.e., we set $G = 0$.

The term $(e_s - e_a)$ in [kPa], is the difference of saturation vapor pressure and actual vapor pressure. The values $e_s$ and $e_a$ are given by

$$e_s = \frac{1}{2}\left(0.6108\, e^{\frac{17.27\, T_{max}}{T_{max}+237.3}} + 0.6108\, e^{\frac{17.27\, T_{min}}{T_{min}+237.3}}\right), \qquad e_a = \frac{RH}{100}\, e_s. \tag{2}$$

The psychrometric constant $\gamma$ depends on the atmospheric pressure $P$ in [kPa] and on above-mentioned constants $C_p = 1013\, \mathrm{J\, kg^{-1}\,{}^\circ C^{-1}}$, $\lambda = 2.45\, \mathrm{MJ\, kg^{-1}}$ and $\varepsilon = 0.622[-]$, with $\varepsilon$ being ratio molecular weight of water vapor to dry air. To compute its value, the following formula is used

$$\gamma = \frac{C_p\, P}{\varepsilon\, \lambda} = 0.665 \times 10^{-3}\, P. \tag{3}$$

Using this formula, the computed value of $\gamma$ depends only on one *measured* quantity and that is *atmospheric* pressure $P$ and is given in $\left[\mathrm{kPa\,{}^\circ C^{-1}}\right]$.

The slope $\Delta$ describes the relationship between saturation vapor pressure and temperature. For a given temperature $T_a$, the corresponding $\Delta$ is given by

$$\Delta = \frac{4098\left(0.6108 e^{\frac{17.27\, T_a}{T_a+237.3}}\right)}{(T_a + 237.3)^2}. \tag{4}$$

The resulting unit of $\Delta$ is $\left[\mathrm{kPa\,{}^\circ C^{-1}}\right]$.

The net radiation at the surface $R_n$ in $\left[\mathrm{MJ\, m^{-2}\, day^{-1}}\right]$ is, by [1], given as the difference incoming net short wave radiation $R_{ns}$ and outgoing net long wave radiation $R_{nl}$, i.e.,

$$R_n = R_{ns} - R_{nl}.$$

To evaluate $R_{nl}$, the knowledge of *solar* radiation $R_s$ and global *extraterrestrial* radiation, $R_a$ is required.

Extraterrestrial radiation is the amount of radiation incident on a unit of the horizontal surface at the outer boundary of the atmosphere. For places of similar latitude, it is approximately the same, changing only during the year. There is no influence of cloud turbidity or air pollution over the Earth's atmosphere, and therefore, the dose of solar energy is the highest at any given time. In addition to the solar constant, the angle of incidence of the sun's rays at a given location of the atmosphere boundary must also be taken into account. Therefore, the value of $R_a$ is expressed depending on these quantities as

$$R_a = \frac{24 \times 60}{\pi}\, G_{sc}\, d_r (\omega_s\, \sin\varphi\, \sin\delta + \sin\omega_s\, \cos\varphi\, \cos\delta). \tag{5}$$

The terms included in Equation (5) are

$G_{sc} = 0.082\, \mathrm{MJ\, m^{-2}\, min^{-1}}$, global solar constant,

$d_r = 1 + 0.33 \cos\left(\frac{2\pi}{365}JD\right)$, Earth-Sun relative distance $[-]$,

$\delta = 0.409 \sin\left(\frac{2\pi}{365}JD - 1.39\right)$, solar declination [rad],

$\varphi$ latitude of the site of interest [rad],

$\omega_s = \arccos\left(-\tan\varphi\, \tan\delta\right)$, sunset hour angle [rad],

$JD$ number of Julian day.

The terms $R_s$ and $R_{ns}$ could be computed by

$$R_s = \left( a_s + b_s \frac{n}{N} \right) R_a, \tag{6}$$

$$R_{ns} = (1 - \alpha) R_s, \tag{7}$$

where $a_s, b_s$ are *Angström* coefficients, $n[\text{h}]$ and $N[\text{h}]$ are *actual* and *maximum possible* duration of daylight, respectively. Finally, $\alpha$ denotes *albedo*, i.e., the coefficient of reflection.

As the *Angström* coefficients are not calculated based on the actual solar radiation measurements here, the *FAO* paper [1] recommendation $a_s = 0.25$ and $b_s = 0.5$ in Equation (6) is used. Furthermore, the *free water surface* albedo is set as $\alpha = 0.08$ based on [1].

The maximum daylight duration $N$ is computed as

$$N = \frac{24}{\pi} \omega_s. \tag{8}$$

To determine *net longwave radiation* $R_{nl}$, the following formula is used

$$R_{nl} = \sigma \left( \frac{(T_{max} + 273.16)^4 + (T_{min} + 273.16)^4}{2} \right) \left( 0.34 - 0.14 \sqrt{e_a} \right) \left( 1.35 \frac{R_s}{R_{so}} - 0.35 \right). \tag{9}$$

The above formula (9) uses the *Stefan-Boltzmann* constant $\sigma = 4.903 \times 10^{-9} \, \text{MJ} \, \text{K}^{-4} \, \text{m}^{-2} \, \text{day}^{-1}$ and $R_{so}$ in $\left[ \text{MJ} \, \text{m}^{-2} \, \text{day}^{-1} \right]$, which is the *clear-sky radiation*.

The value of the *clear-sky radiation* $R_{so}$ is calculated as

$$R_{so} = \left( 0.75 + 2 \times 10^{-5} z \right) R_a,$$

where $z$ is the site *altitude* in $[\text{m}]$ above the sea level.

However, despite its complexity of the FAO Penman–Monteith Equation (1), it is not possible to consider $E_{FAO}$ results to be accurate, since the number of input data to be measured or calculated by empirical formulae based on measured input data. Such an estimation process is affected by measurement and calculation errors. For example, in [32,33], one could find sensitivity analysis of the FAO Penman–Monteith equation in different climate conditions.

It should be pointed out that Equation (1) was derived as a method to determine the *reference rate of evapotranspiration*, i.e., the reference rate of evaporation from growing plants with the characteristics of *hypothetical reference crops* such as *height*, *aerodynamic resistance* of their surface, and *albedo*. For *real crops*, the rate of evapotranspiration is determined from $E_{FAO}$ by multiplying the crop-related coefficient $K_c$

$$E_{crop} = K_c E_{FAO}.$$

With the proper coefficient, the $E_{FAO}$ formula could be used to estimate open water evaporation. The values of coefficient $K_c$, i.e., $K_{c,mid}$ and $K_{c,end}$ for mid and end season respectively are tabulated in [1]. Specifically, $K_{c,mid} = K_{c,end} = 1.05$ for *shallow* lakes, i.e., for those with a depth of up to 2 m. For *deep* lakes, i.e., with a depth exceeding 5 m, the values $K_{c,mid} = 0.65$ and $K_{c,end} = 1.25$ are indicated. Therefore, it should be borne in mind that (especially in the case of deep lakes) the result of $E_{FAO}$ could lead to the underestimation of up to 35% or the overestimation up to 25% during the season.

Since our research is not focused only on the area of Lake Most, but also on lakes that are only planned and does not exist at present, it is impossible to use limnological and bathymetric data, such as temperature profile or water depth. This leads us to the considerations of the article [7], which states that in the case of missing *limnological* data, the lake coefficient $K_c = 1$ can be selected and $E_{FAO}$ result itself could be considered to be

an open water evaporation estimate. Hence, in all our calibration and validation processes, the equation $E_{FAO}$ serves us as an etalon and all our results are compared against it.

*2.4. Evaporation Estimation Methods*

Since the FAO Penman–Monteith equation $E_{FAO}$ (see (1) in Section 2.3) is very input-intensive and complex in its calculation procedure, many other methods have been derived to determine the rate of evaporation. Depending on the inputs of the method, we divide them into temperature-, radiation-, mass transfer-based, and combined methods.

Temperature-based method equations can be considered the simplest type of equations. They primarily work with a single variable, namely the average *mean air temperature $T_a$*. Quite often these equations have a linear form $E = p\,T_a + q$, but they also occur in the form $E = k\,T_a^m$, or the form of exponential formula $E = 10^{p\,T_a+q}$ or $E = e^{p\,T_a+q}$. However, the group also includes relations in which the temperature occurs in combination with a member comprising, for instance, *relative humidity RH* or *theoretical length of the solar* day $N$.

In this section, we selected 7 simple (in comparison to the complexity of FAO Penman–Monteith) evaporation models for the demonstration of our calibration approach.

2.4.1. Regression Derived Relations—Czech Republic

The following three relations are used in the Czech Republic. They are derived by regression between the observed evaporation and mean daily air temperature, using statistical regression to find both *linear* and *exponential* models. The model relations presented in this section are compared in the paper [34] with measurements on $20\,\text{m}^2$ evaporation pan placed in the meteorological station *Hlasivo* near the city of Tábor $49°\text{C}290'$ N, $14°\text{C}450'$ E) in the South Bohemian Region. This station is operated by *Výzkumný ústav vodohospodářský T. G. Masaryka* (VUV, T. G. Masaryk Water Research Institute) and was built in 1957 and has a $20\,\text{m}^2$ evaporation tank, *GGI*-3000 pan and *Class-A* pan. The pan evaporation measurements here are carried out from May to October, which is due to the temperatures below the freezing point in the winter months. The models are given by

$$E_{\text{S}} = 10^{0.0452\,T_a - 0.204}, \tag{10}$$

$$E_{\text{BV}} = 0.2157\,T_a + 0.1133, \tag{11}$$

$$E_{\text{VUV}} = 0.2157\,T_a + 0.726\,u_2 - 1.2259, \tag{12}$$

where $E_S$ is the equation according to Šermer [35], $E_{BV}$ according to Beran and Vizina [36], and $E_{VUV}$ according to Adam Beran from VUV published in the official report *Model průběhu meteorologických veličin pro oblast jezera Most do roku 2050* (The modelling of the course of meteorological quantities for Lake Most area until 2050). In all equations, the evaporation rate is determined in $[\text{mm day}^{-1}]$. Equation (10) has the form of an exponential function and therefore its results can never be negative. However, it must be mentioned that there are limitations of Equations (11) and (12): if the equation produces the negative evaporation estimation, we set the value equal to zero. For instance, $E_{BV} = 0$ is set on days with the mean temperature below $-0.526\,°\text{C}$, since $E_{BV}$ would be negative in such cases.

To calibrate the models, we present a parametric formulation of the Equations (10)–(12) by

$$E_{\text{S}}(\theta) = 10^{\theta_1 T_a + \theta_2}, \tag{13}$$

$$E_{\text{BV}}(\theta) = \max\{\theta_1 T_a + \theta_2, 0\}, \tag{14}$$

$$E_{\text{VUV}}(\theta) = \max\{\theta_1 T_a + \theta_2 u_2 + \theta_3, 0\}, \tag{15}$$

where $\theta$ are unknown parameters, which will be calibrated. We extended models by projection to nonnegative numbers (using the outer max function) to enforce the computed nonnegative evaporation.

### 2.4.2. Kharrufa

The equation presented by *Kharrufa* in [37] is an example of a nonlinear temperature formula. It is mostly written in the literature in the form

$$E_{\mathrm{K}} = 0.34 \, p \, T_a^{1.3} \tag{16}$$

and results in the evaporation rate in $[\mathrm{mm \, day^{-1}}]$. In (16), variable $p$ denotes the percentage of total daytime hours for the daily period out of total daytime hours of the year. This form is used, for instance, in [38,39]. The coefficient 0.34 was found empirically and it is possible to refine it with respect to the site-specific conditions. For example, in [34], the form $E_K = 0.25 \, p \, T_a^{1.3}$ is given with the formula being calibrated for the conditions of the *Hlasivo* weather station in the South Bohemian Region. In this study, the form (16) with coefficient 0.34 is used.

In this paper, we the calibrate model (16) introducing the parametric version

$$E_{\mathrm{K}}(\theta) = \begin{cases} \theta_1 p T_a^{\theta_2} & \text{if } T_a > 0, \\ 0 & \text{if } T_a \leq 0, \end{cases} \tag{17}$$

and calibrate the unknown parameters $\theta \in \mathbb{R}^2$.

### 2.4.3. Hargreaves–Samani

Another method was introduced by *Hargreaves* in article [40] and further modified to the form which can be found in article [41]. Usually, the *Hargreaves–Samani* equation is given in its basic form

$$E_{HS} = 0.0023 \, R_a \, T_r^{\frac{1}{2}} \, (T_a + 17.8).$$

In this equation, variable $T_r$ denotes the difference between daily maximum and minimum air temperatures $[^{\circ}\mathrm{C}]$.

Although the formula contains a radiation term $R_a$, it is ranked among the temperature-based formulae since the term $R_a$ here is just a theoretical value calculated according to Formula (5). Using this computation of $R_a$, the *Hargreaves–Samani* equation takes any of the following equivalent forms

$$E_{HS} = 0.0023 \, \frac{R_a}{\lambda}(T_a + 17.8)\sqrt{T_r} = 0.0023(0.408 \, R_a)(T_a + 17.8)\sqrt{T_r} = 0.00094 \, R_a(T_a + 17.8)\sqrt{T_r}. \tag{18}$$

The difference of these forms is only in the usage of division by the *latent heat of vaporization* of water $\lambda = 2.45 \, \mathrm{MJ \, kg^{-1}}$, which is performed to obtain the results in millimetres per day.

In this paper, we consider the parametric form of Hargreaves–Samani Equation (18)

$$E_{\mathrm{HS}}(\theta) = \max\left\{ \frac{\theta_1 R_a(T_a + \theta_2)\sqrt{T_r}}{\lambda}, 0 \right\} \tag{19}$$

with parameters $\theta \in \mathbb{R}^2$. These parameters will be optimized during the calibration process.

### 2.4.4. Schendel

In contrast to the formulae mentioned above, in which the air temperature is sufficient to calculate the evaporation rate, the air relative humidity $RH$ measurement is required in the following *Schendel* equation. The formula has a simple form

$$E_{Sch} = 16 \, \frac{T_a}{RH}. \tag{20}$$

The equation can be found in the original *Schendel* paper [42]. It is used by many authors, for example, see [19,20].

To calibrate this model, we consider the Schendel Equation (22) in the parametric form

$$E_{\text{Sch}}(\theta) = \max\left\{\theta_1 \frac{T_a}{RH}, 0\right\} \tag{21}$$

with unknown parameter $\theta \in \mathbb{R}$.

### 2.4.5. Priestley–Taylor Equation

A potential evapotranspiration based on the Priestley–Taylor can be considered to be a combined method, which is developed as a combination of the turbulent diffusion method and the method of energy balance [43]. The basic equation for the computation of the potential evapotranspiration by the Priestley–Taylor method is given by

$$E_{\text{PT}} = 1.26 \frac{\Delta \cdot R_n}{\lambda}. \tag{22}$$

Please see Section Abbreviations in the end of this paper for the description of the used variables.

In this paper, we optimize the constant parameter in Equation (22), i.e., we consider a parametric model

$$E_{\text{PT}}(\theta) = \max\left\{\theta \frac{\Delta \cdot R_n}{\lambda}, 0\right\}, \tag{23}$$

where $\theta \in \mathbb{R}$ is unknown parameter.

### 2.4.6. Turc Equation

The Turc method was developed for the climatic conditions of western Europe [44]. Several forms of Turc equation could be found in the literature. In this paper, we are considering the one from [45], i.e.,

$$E_{\text{T}} = 0.0133 \, \frac{T_a}{T_a + 15} \, (23.8856 \, R_s + 50) \, C_{RH},$$

$$\text{with} \quad C_{RH} = \begin{cases} 1 & \text{for} \quad RH \geq 50, \\ 1 + \frac{50 - RH}{70} & \text{for} \quad RH < 50. \end{cases} \tag{24}$$

The term $R_s$ *total solar radiation* is computed using Equation (6).

In this paper, we calibrate the Turc Equation (24) with respect to two parameters, namely we work with the parametric model

$$E_{\text{T}}(\theta) = \max\left\{\theta_1 \, \frac{T_a}{T_a + 15} \, (23.8856 \, R_s + \theta_2) \, C_{RH}, 0\right\}, \tag{25}$$

where the term $C_{RH}$ and its conditions remains the same as in (24).

### 2.5. Statistical Measures

To compare the performance and predictive power of each evaporation estimate method $E$ defined in the previous section against the FAO Penman–Monteith equation, the following statistical measures are used: *Nash–Sutcliffe efficiency* (NSE), *root mean square error* (RMSE), *mean absolute error* (MAE), and *percent bias* (PBIAS). Each of these measures offers a description of the difference between *observed or measured* and *predicted or calculated* values.

One of the most popular measures that assesses model predictive power is *Nash–Sutcliffe efficiency* (NSE), see [46], which tries to capture the extent of errors and their degree of variability. It is given by

$$\text{NSE}(E_{\text{FAO}}, E) = 1 - \left[ \frac{\sum\limits_{t=1}^{T} (E_{\text{FAO}}\langle t \rangle - E\langle t \rangle)^2}{\sum\limits_{t=1}^{T} (E_{\text{FAO}}\langle t \rangle - \bar{E}_{\text{FAO}})^2} \right], \tag{26}$$

where $T$ is several observations and $\bar{E}_{\text{FAO}}$ is the mean value of FAO Penman–Monteith model given by

$$\bar{E}_{\text{FAO}} = \frac{1}{T} \sum_{t=1}^{T} \bar{E}_{\text{FAO}}\langle t \rangle.$$

The NSE index indicates the relative magnitude of the residual variance ("noise") compared to the measured data variance ("information"), see [27]. The index shows, among other things, how well the *scatterplot* of the observed and modelled data corresponds to a 1:1 straight line. NSE takes the value $-\infty \leq \text{NSE} \leq 1$ and NSE $= 1$ indicates a perfect match. The value NSE $= 0$ means that the model predicts with the same accuracy as the observation mean. Negative values of NSE indicate an *unacceptable* model.

To capture the size of individual errors $E\langle t \rangle - E_{\text{FAO}}\langle t \rangle$, *root mean square error* (RMSE) and the *mean absolute error* (MAE) could be used. They are defined by

$$\text{RMSE}(E_{\text{FAO}}, E) = \sqrt{\frac{1}{T} \sum_{t=1}^{T} (E\langle t \rangle - E_{\text{FAO}}\langle t \rangle)^2}, \tag{27}$$

$$\text{MAE}(E_{\text{FAO}}, E) = \frac{1}{T} \sum_{t=1}^{T} |E\langle t \rangle - E_{\text{FAO}}\langle t \rangle|. \tag{28}$$

Both RMSE and MAE also express the model error in units corresponding to the units of the value observed. The MAE value captures the mean error size and similar information is provided by RMSE. However, RMSE attaches more weight to larger errors and thus suggests the presence of "extreme error". For both MAE and RMSE, it should be noted that their results are always greater than zero, which results in the loss of overvaluation or undervaluation information. For both, their lower value indicates a better model. Concerning the MAE and RMSE indicators, the "half standard deviation" rule is also mentioned; i.e., in a good model, MAE, as well as RMSE should be less than half the standard deviation of the observed variable.

The simplest measure of the prediction quality is the difference $E_{\text{FAO}} - E$ or the percentage expression of this difference. This value is called *percent bias* and it is computed as

$$\text{PBIAS}(E_{\text{FAO}}, E) = \frac{\sum_{t=1}^{T}(E\langle t \rangle - E_{\text{FAO}}\langle t \rangle)}{\sum_{t=1}^{T} E_{\text{FAO}}\langle t \rangle}, \tag{29}$$

where $E\langle t \rangle$ is the value of the model in time (day) $t = 1, \ldots, T$.

We used the presented statistical measures to calibrate the parameters of the models, i.e., for each model and measure, we solve the minimization problem

$$\theta^* = \arg\min \rho(E_{\text{FAO}}, E(\theta)) \tag{30}$$

in the case of $\rho \in \{\text{RMSE}, \text{MAE}, \text{PBIAS}\}$ (given by Equations (27)–(29)), or the maximization problem

$$\theta^* = \arg\max \rho(E_{\text{FAO}}, E(\theta)) = \arg\min -\rho(E_{\text{FAO}}, E(\theta)) \tag{31}$$

in the case of $\rho = \text{NSE}$ (given by (26)). In both problems, $E(\theta)$ represents the considered parametric model which has to be calibrated, i.e., one of Equations (13)–(15), (17), (19), (21), or (23).

During our experiments on model calibration using different statistical measures, we observed that the maximization of NSE and minimization of RMSE produce the same optimizer. The following theorem supports this observation with theoretical proof.

**Theorem 1.**
$$\arg\max_{\theta} \; \mathrm{NSE}(E_{\mathrm{FAO}}, E(\theta)) = \arg\min_{\theta} \; \mathrm{RMSE}(E_{\mathrm{FAO}}, E(\theta)) \tag{32}$$

**Proof.** From the definition of the optimality point $\theta^*$ of the optimization problem on the left side of (32), we can write for all possible parameters $\theta$

$$\mathrm{NSE}(E_{\mathrm{FAO}}, E(\theta)) \leq \mathrm{NSE}(E_{\mathrm{FAO}}, E(\theta^*)),$$

i.e., using the definition (26)

$$1 - \left[ \frac{\sum\limits_{t=1}^{T} (E_{\mathrm{FAO}}\langle t\rangle - E(\theta)\langle t\rangle)^2}{\sum\limits_{t=1}^{T} (E_{\mathrm{FAO}}\langle t\rangle - \bar{E}_{\mathrm{FAO}})^2} \right] \leq 1 - \left[ \frac{\sum\limits_{t=1}^{T} (E_{\mathrm{FAO}}\langle t\rangle - E(\theta^*)\langle t\rangle)^2}{\sum\limits_{t=1}^{T} (E_{\mathrm{FAO}}\langle t\rangle - \bar{E}_{\mathrm{FAO}})^2} \right].$$

This inequality can be modified to an equivalent form subtracting 1 from both sides and multiplying by constant negative term $-\sum\limits_{t=1}^{T} (E_{\mathrm{FAO}}\langle t\rangle - \bar{E}_{\mathrm{FAO}})^2$ to obtain

$$\sum_{t=1}^{T} (E_{\mathrm{FAO}}\langle t\rangle - E(\theta)\langle t\rangle)^2 \geq \sum_{t=1}^{T} (E_{\mathrm{FAO}}\langle t\rangle - E(\theta^*)\langle t\rangle)^2.$$

We divide this inequality by a positive number of data points $T$ and apply the square root to both sides. Since both sides of the original inequality are nonnegative values and the square root is an increasing function, the following inequality is equivalent to the previous one

$$\sqrt{\frac{1}{T} \sum_{t=1}^{T} (E_{\mathrm{FAO}}\langle t\rangle - E(\theta)\langle t\rangle)^2} \geq \sqrt{\frac{1}{T} \sum_{t=1}^{T} (E_{\mathrm{FAO}}\langle t\rangle - E(\theta^*)\langle t\rangle)^2},$$

i.e., using the definition (27)

$$\mathrm{RMSE}(E_{\mathrm{FAO}}, E(\theta)) \leq \mathrm{RMSE}(E_{\mathrm{FAO}}, E(\theta^*)).$$

This is the optimality condition for the solution of the optimization problem on the right side of (32).  □

### 2.6. Implementation

We have implemented the calibration process in R programming language [47]. This software provides us with an easy way how to load the data, manipulate them, solve the corresponding optimization problems, and present the results.

The type of the optimization problem depends on the considered model and the selected statistical measure, see (30) and (31). We compared several solvers provided by R programming language (both in computational time and the accuracy of the results) and we observed that the *nlminb* algorithm from *optim* package is the most efficient option for solving the optimization problems. This algorithm is using PORT routines [48] and our numerical experiments proved the suitability and stability during the solution of both linear and nonlinear models.

To generalize our calibration process, we adopt the *cross-validation* methodology [28]. The modelling with a random subset of data generalizes the results and avoids overfit-

ting. In our case, we perform 10 random permutations of the data (please notice that the statistical measures which we are using are independent of the order in time). We split each permutation into 10 parts—9 of them is used for calibrating the model (i.e., solution of (30) or (31)) and the remaining part is used for validation. See Figure 3, where we demonstrate performed 10 calibration-validation processes on one specific data permutation. We repeat this permutation 100 times and for each of this permutation, we repeat 10 calibration-validation splittings. In total, we obtain 1000 results of the calibration process, which are further analyzed. This method is well known as K-fold cross-validation.

**Figure 3.** The cross-validation: one random permutation of input data is divided into 10 parts, 9 of them are used for calibration, the remaining one is used for validation. On one specific data permutation, we obtain 10 different calibration results. In this figure, we demonstrate the process on the data of length $T = 20$; however, our real dataset is of length $T = 2191$.

## 3. Results

We calibrate the models (Section 2.4) against the FAO Penman–Monteith equation (Section 2.3) using the criteria (Section 2.5) and cross-validation process (Section 2.6). In the comparison of the original models, the calibration process always results in better statistical values, see Table 1. The value has been obtained using the parameters presented in Table 2 on the whole data set.

**Table 1.** The comparison of statistical measures before and after the calibration process. These values have been computed on the whole data set and correspond to the models presented in Table 2.

| | NSE | | RMSE | | MAE | | PBIAS | |
|---|---|---|---|---|---|---|---|---|
| | **Original** | **Calibrated** | **Original** | **Calibrated** | **Original** | **Calibrated** | **Original** | **Calibrated** |
| $E_{\text{S}}$ | 0.64947 | 0.76350 | 1.06670 | 0.87618 | 0.78435 | 0.69236 | 13.53217 | −0.00762 |
| $E_{\text{BV}}$ | 0.75216 | 0.76917 | 0.89694 | 0.86561 | 0.70983 | 0.69185 | 7.97212 | 0.01534 |
| $E_{\text{VUV}}$ | 0.69760 | 0.78173 | 0.99077 | 0.84180 | 0.76077 | 0.67377 | −1.68248 | 0.01862 |
| $E_{\text{K}}$ | 0.66759 | 0.85119 | 1.03877 | 0.69500 | 0.77958 | 0.54785 | 15.48084 | 0.00038 |
| $E_{\text{HS}}$ | 0.89358 | 0.93006 | 0.58776 | 0.47650 | 0.43779 | 0.34324 | 10.93232 | 0.00307 |
| $E_{\text{Sch}}$ | 0.82408 | 0.86145 | 0.75568 | 0.67059 | 0.58732 | 0.52887 | 10.94825 | −0.00027 |
| $E_{\text{PT}}$ | 0.91229 | 0.97519 | 0.53359 | 0.28381 | 0.40145 | 0.19360 | 11.75804 | −0.00316 |
| $E_{\text{T}}$ | 0.93357 | 0.96395 | 0.46437 | 0.34210 | 0.37318 | 0.26877 | -14.98293 | 0.00117 |

**Table 2.** The comparison of original and calibrated models. The calibrated models have been obtained as a mean value of the cross-validation process. The presented models have the values of statistical measures corresponding to Table 1. Despite the theoretical equality of NSE and RMSE optimal parameters (see Theorem 1), the numerical algorithm provides slightly different solutions.

| Original | NSE | RMSE | MAE | PBIAS |
|---|---|---|---|---|
| $E_S = 10^{0.0452 T_a - 0.204}$ | $E_S = 10^{0.0372 T_a - 0.1224}$ | $E_S = 10^{0.0373 T_a - 0.1255}$ | $E_S = 10^{0.0422 T_a - 0.2312}$ | $E_S = 10^{0.0417 T_a - 0.2025}$ |
| $E_{BV} = 0.2157 T_a + 1.1133$ | $E_{BV} = 0.2325 T_a - 0.356$ | $E_{BV} = 0.2327 T_a - 0.3578$ | $E_{BV} = 0.2265 T_a - 0.2669$ | $E_{BV} = 0.1992 T_a + 0.1119$ |
| $E_{VUV} = 0.2157 T_a + 0.726 u_2 - 1.2259$ | $E_{VUV} = 0.2445 T_a + 0.2365 u_2 - 0.8668$ | $E_{VUV} = 0.2447 T_a + 0.2243 u_2 - 0.8498$ | $E_{VUV} = 0.2386 T_a + 0.2672 u_2 - 0.8987$ | $E_{VUV} = 0.2192 T_a + 0.7264 u_2 - 1.2253$ |
| $E_K = 0.34 p T_a^{1.3}$ | $E_K = 0.8508 p T_a^{0.9086}$ | $E_K = 0.8854 p T_a^{0.894}$ | $E_K = 0.8184 p T_a^{0.9182}$ | $E_K = 0.3358 p T_a^{1.2526}$ |
| $E_{HS} = \frac{0.0023 R_a (T_a + 17.8) \sqrt{T_t}}{\lambda}$ | $E_{HS} = \frac{0.0021 R_a (T_a + 17.5571) \sqrt{T_t}}{\lambda}$ | $E_{HS} = \frac{0.0021 R_a (T_a + 17.5665) \sqrt{T_t}}{\lambda}$ | $E_{HS} = \frac{0.0021 R_a (T_a + 16.2065) \sqrt{T_t}}{\lambda}$ | $E_{HS} = \frac{0.0021 R_a (T_a + 17.8) \sqrt{T_t}}{\lambda}$ |
| $E_{Sch} = 16 \frac{T_a}{RH}$ | $E_{Sch} = 14.2663 \frac{T_a}{RH}$ | $E_{Sch} = 14.1976 \frac{T_a}{RH}$ | $E_{Sch} = 14.2161 \frac{T_a}{RH}$ | $E_{Sch} = 14.4211 \frac{T_a}{RH}$ |
| $E_{PT} = 1.26 \frac{\Delta (R_n - G)}{\lambda (\Delta + \gamma)}$ | $E_{PT} = 1.0876 \frac{\Delta (R_n - G)}{\lambda (\Delta + \gamma)}$ | $E_{PT} = 1.0876 \frac{\Delta (R_n - G)}{\lambda (\Delta + \gamma)}$ | $E_{PT} = 1.0714 \frac{\Delta (R_n - G)}{\lambda (\Delta + \gamma)}$ | $E_{PT} = 1.1274 \frac{\Delta (R_n - G)}{\lambda (\Delta + \gamma)}$ |
| $E_T = 0.0133 \frac{T_a}{T_a + 15} (23.8856 R_s + 50) C_{RH}$ | $E_T = 0.0141 \frac{T_a}{T_a + 15} (23.8856 R_s + 78) C_{RH}$ | $E_T = 0.0141 \frac{T_a}{T_a + 15} (23.8856 R_s + 78) C_{RH}$ | $E_T = 0.0148 \frac{T_a}{T_a + 15} (23.8856 R_s + 50) C_{RH}$ | $E_T = 0.0156 \frac{T_a}{T_a + 15} (23.8856 R_s + 50) C_{RH}$ |

The parameters of the optimal models have been chosen as a mean value of the statistical measure on testing data. As was described in Section 2.6, we are calibrating models on the random part of a given data set and since the optimal parameters depend on this choice, the calibrated parameters are also random variables. Consequently, the new value of the statistical measure is random as well. See Figure 4, where we demonstrate the *randomness* of the statistical value—we plot the statistical measure of the original model on different training data and compare them with the statistical measures obtained by the calibration process on these training data. The final calibrated model has been chosen as a model, which corresponds to the mean value of the obtained calibrations. Each of the presented graphs corresponds to different statistical measures in the calibration.

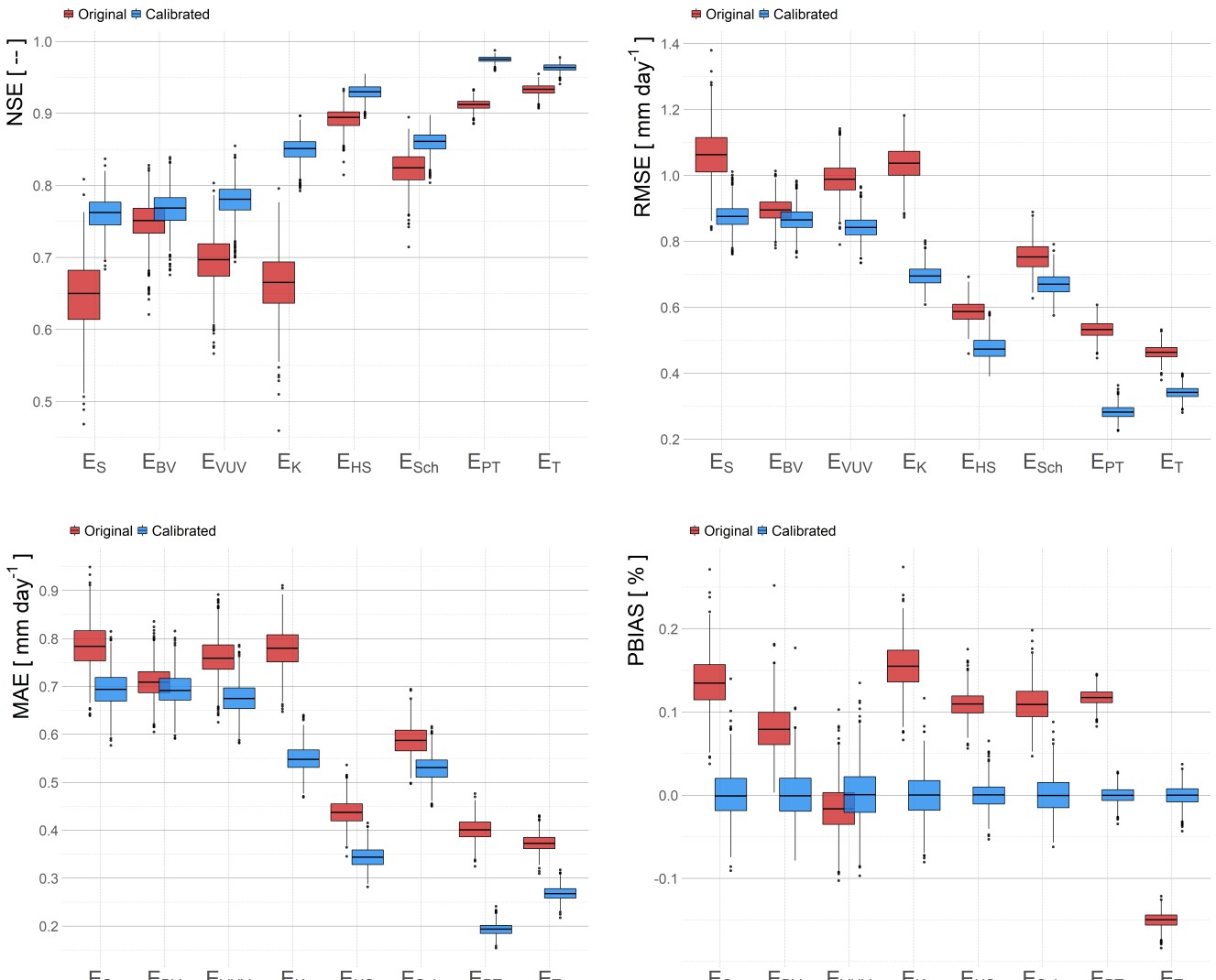

**Figure 4.** The comparison of the statistical measure of the original model and the calibrated model on the validation part of the data. From these results, the optimal calibrated model is the one that corresponds to the mean value of the calibrated statistical measures.

We calibrated the models with respect to all considered statistical measures and we track the change of other measures. As an example, see Tables 3–5 for the results in the case of the Kharrufa model, Hargreaves–Samani model, and Turc model.

**Table 3.** The comparison of statistical measures before and after the calibration process of the Kharrufa equation. The first row represents the values of various statistical measures (see columns) of the original model. Remaining rows determine the objective statistical measure, which respect to the model was calibrated. The columns represent the corresponding values of various measures.

| | | Obtained Value | | | |
|---|---|---|---|---|---|
| | | **NSE** | **RMSE** | **MAE** | **PBIAS** |
| Calibration objective | Original | 0.6675858 | 1.0387729 | 0.7795765 | 15.480837 |
| | NSE | 0.8511901 | 0.6950192 | 0.5481185 | −1.191246 |
| | RMSE | 0.8511973 | 0.6950022 | 0.5482104 | −1.156551 |
| | MAE | 0.8509640 | 0.6955467 | 0.5478520 | −2.451671 |
| | PBIAS | 0.8149230 | 0.7750987 | 0.6029666 | 0.003074 |

**Table 4.** The comparison of statistical measures before and after the calibration process of the Hargreaves–Samani equation. The first row represents the values of various statistical measures (see columns) of the original model. Remaining rows determine the objective statistical measure, which respect to the model was calibrated. The columns represent the corresponding values of various measures.

| | | Obtained Value | | | |
|---|---|---|---|---|---|
| | | **NSE** | **RMSE** | **MAE** | **PBIAS** |
| Calibration objective | Original | 0.8935767 | 0.5877587 | 0.4377886 | 10.9323242 |
| | NSE | 0.9300600 | 0.4764787 | 0.3440658 | −1.2411942 |
| | RMSE | 0.9300550 | 0.4764958 | 0.3442857 | −1.0261792 |
| | MAE | 0.9298460 | 0.4772072 | 0.3432423 | −2.2165424 |
| | PBIAS | 0.9297419 | 0.4775611 | 0.3461323 | 0.0023911 |

**Table 5.** The comparison of statistical measures before and after the calibration process of the Turc equation. The first row represents the values of various statistical measures (see columns) of the original model. Remaining rows determine the objective statistical measure, which respect to the model was calibrated. The columns represent the corresponding values of various measures.

| | | Obtained Value | | | |
|---|---|---|---|---|---|
| | | **NSE** | **RMSE** | **MAE** | **PBIAS** |
| Calibration objective | Original | 0.6675858 | 1.03877290 | 0.7795765 | 15.4808373 |
| | NSE | 0.8511901 | 0.6950192 | 0.5481185 | −1.1912456 |
| | RMSE | 0.8511973 | 0.6950022 | 0.5482104 | −1.1565508 |
| | MAE | 0.8509640 | 0.6955467 | 0.5478520 | −2.4516705 |
| | PBIAS | 0.8149230 | 0.7750987 | 0.6029666 | 0.0030740 |

As we mentioned above, the random choice of calibration data in the calibration process causes the randomness of the optimal parameters. In Figures 5–7, we present the different optimal values of calibrated models with respect to various statistical measures. We can observe that using the cross-validation approach, the final optimal parameters are a random variable as well.

The calibration process fits the model to the values computed by the FAO Penman–Monteith equation. We demonstrate this capability on specific examples. Figures 8–10 present the comparison of monthly evaporation computed by FAO Penman–Monteith Equation (1), Hargreaves–Samani Equation (16), and calibrated Hargreaves–Samani Equation (17) in the form of cumulative sum and the scatter plot. The calibration has been performed with respect to NSE.

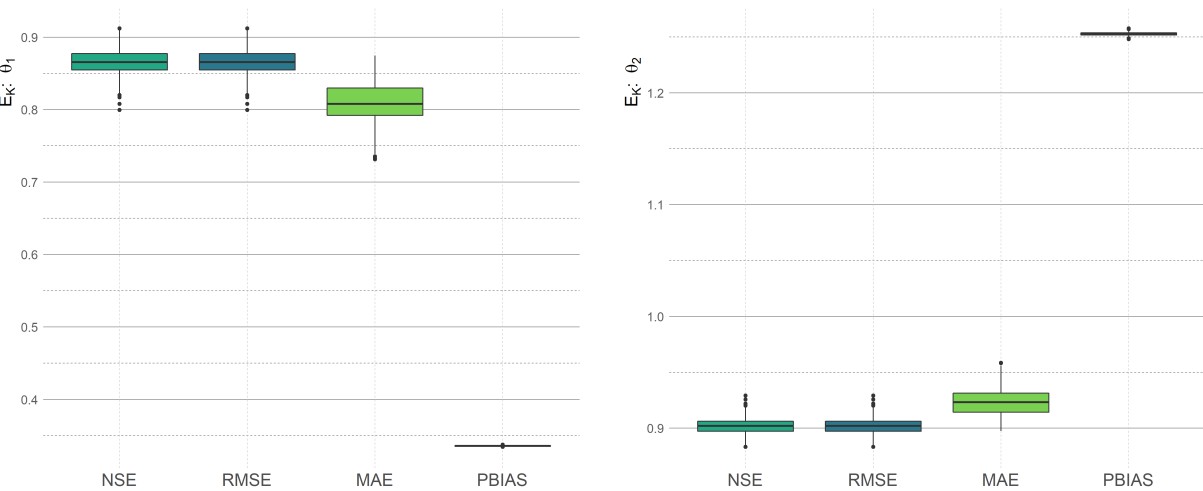

**Figure 5.** The comparison of the calibrated Kharrufa model parameters obtained by the cross-validation calibration process on training data with respect to various statistical measures.

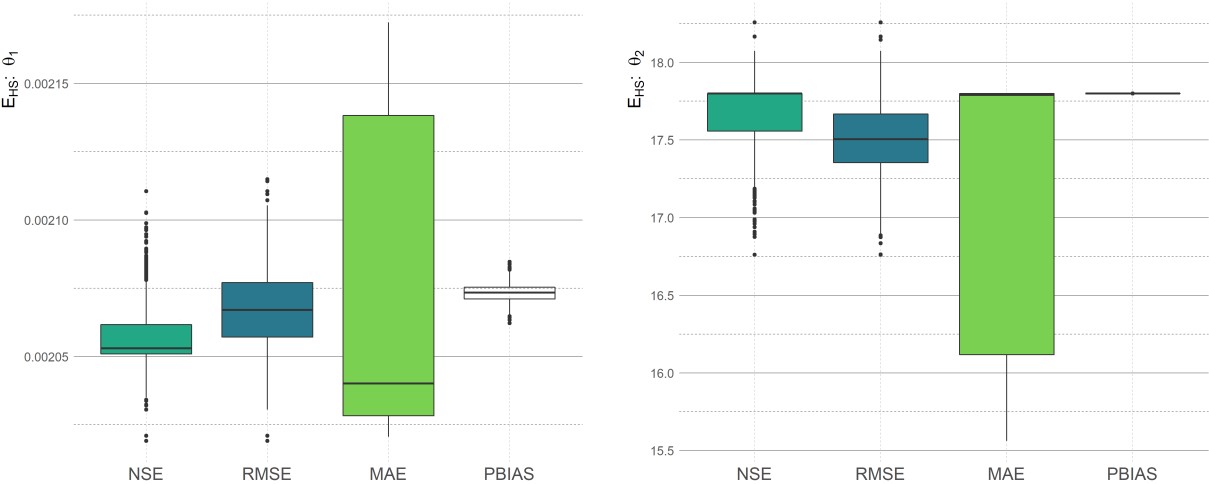

**Figure 6.** The comparison of the calibrated Hargreaves–Samani model parameters obtained by the cross-validation calibration process on training data with respect to various statistical measures.

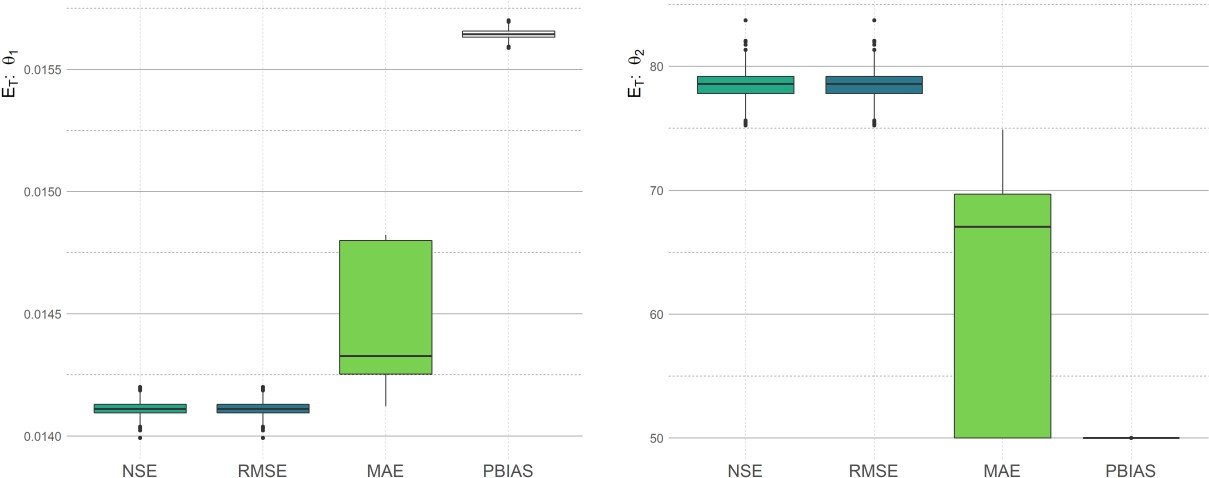

**Figure 7.** The comparison of the calibrated Turc model parameters obtained by the cross-validation calibration process on training data with respect to various statistical measures.

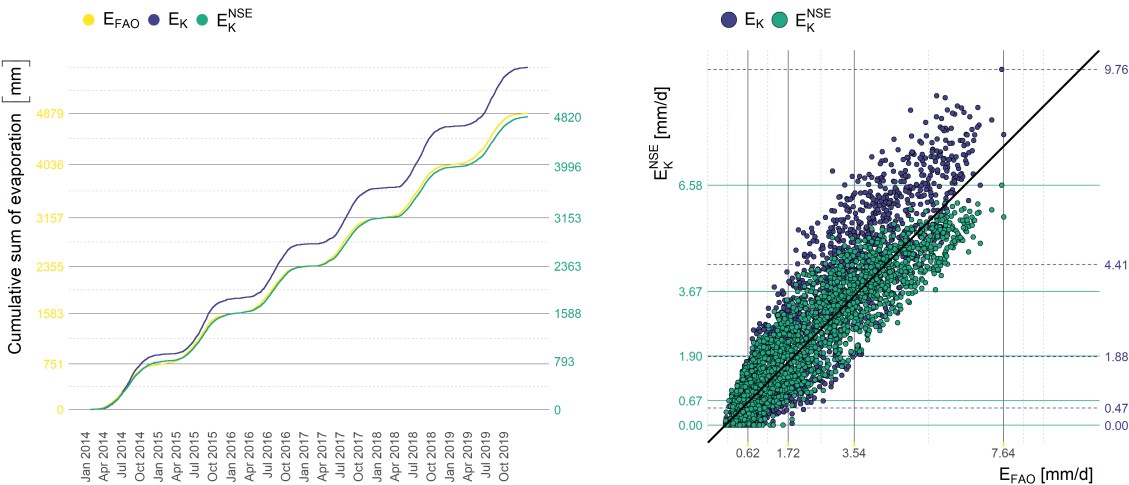

**Figure 8.** The comparison of the daily evaporation computed by FAO Penman–Monteith equation, Kharrufa equation, and calibrated Kharrufa equation with respect to NSE in the form of cumulative evaporation (**left**) and the scatter plot (**right**).

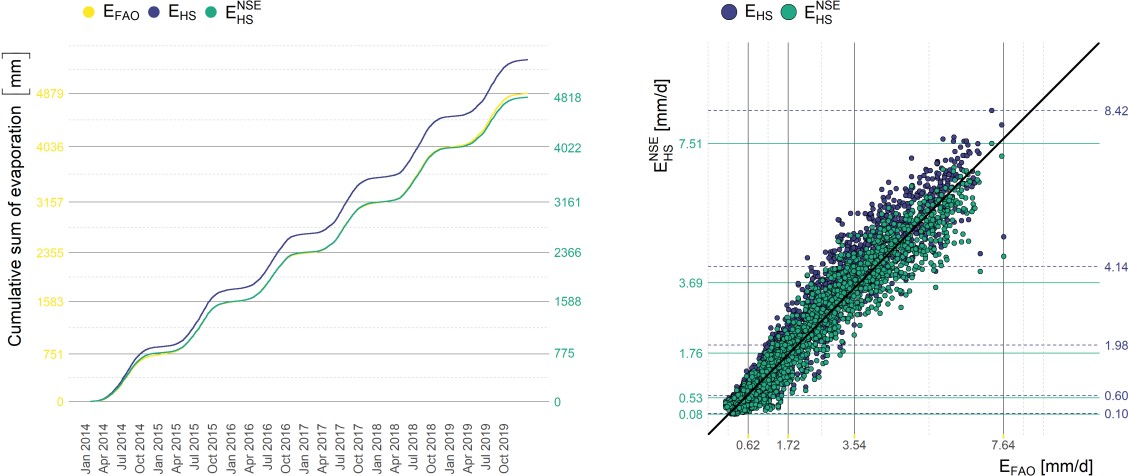

**Figure 9.** The comparison of the daily evaporation computed by FAO Penman–Monteith equation, Hargreaves–Samani equation, and calibrated Hargreaves–Samani equation with respect to NSE in the form of cumulative evaporation (**left**) and the scatter plot (**right**).

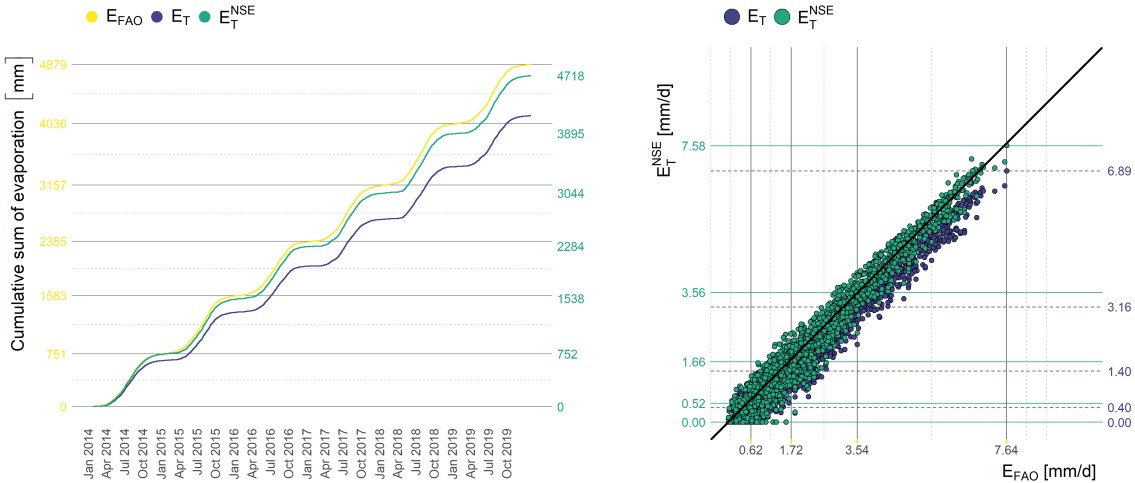

**Figure 10.** The comparison of the daily evaporation computed by FAO Penman–Monteith equation, Turc equation, and calibrated Turc equation with respect to NSE in the form of cumulative evaporation (**left**) and the scatter plot (**right**).

The final result is presented in Figure 11. Here, we demonstrate the monthly evaporation computed by Hargreaves–Samani equation in the form of a time-line and histogram of evaporation in months. The parameters of the calibrated models can be found in Table 2 and the improvement of the used statistical measure on the whole data set in Table 1.

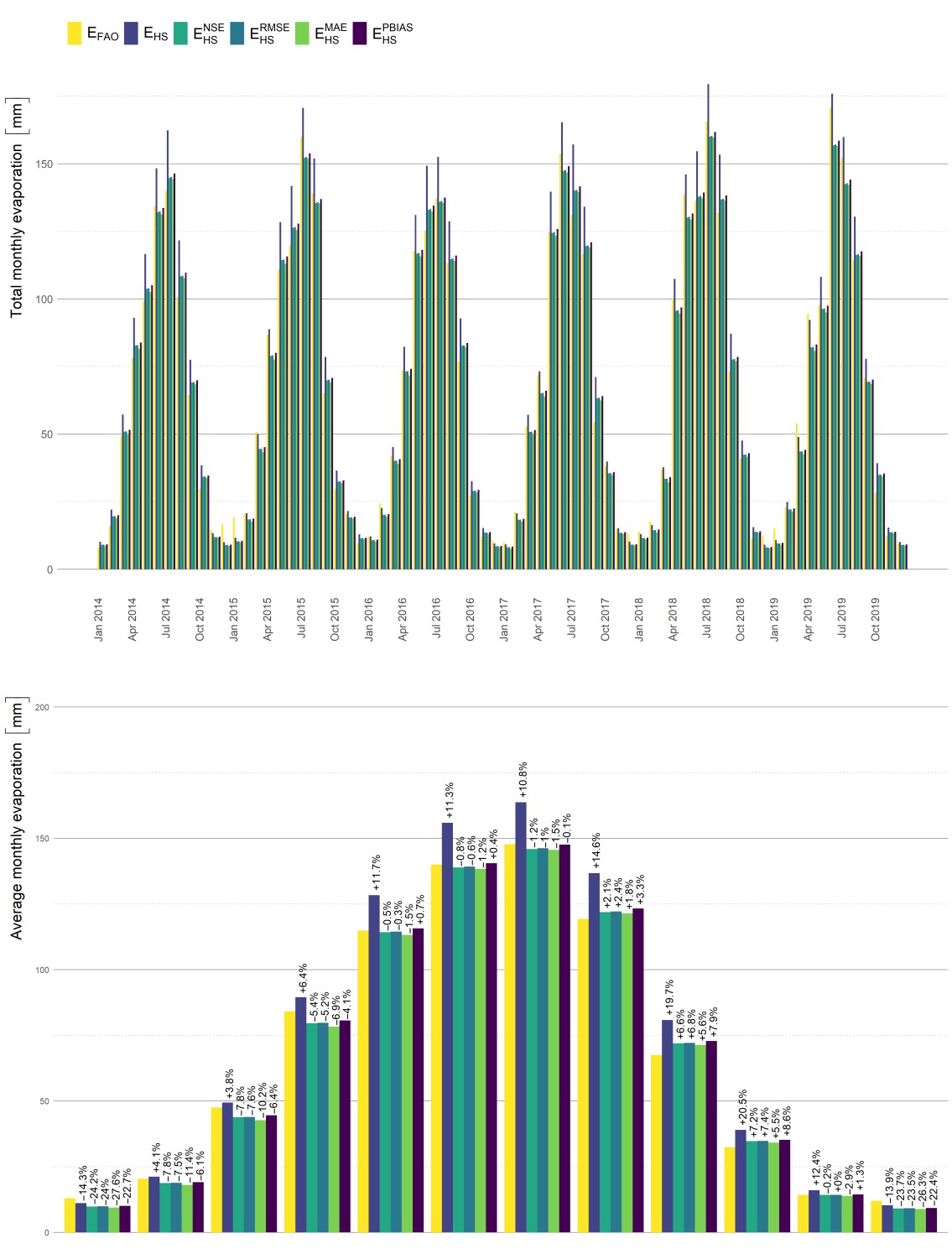

**Figure 11.** The comparison of the monthly evaporation computed by FAO Penman–Monteith equation, Hargreaves–Samani equation, and calibrated Hargreaves–Samani equation. The calibration has been performed using various stochastic measures.

## 4. Discussion

In our results presented in the previous section, we processed the results of calibration on random data from the cross-validation process (see Section 2.6). Before the selection of the mean model, we removed the outliers based on the quartile thresholding. For instance, in the case of the Kharrufa equation calibrated with respect to *NSE*, we removed 11 outliers more than 1.5 interquartile ranges (IQRs) below the first quartile or above the third quartile.

Our results show that the calibrated parameters depend on the chosen statistical measure, see Table 2. However, all of them are improving the objective value in comparison with the original equations, see Table 1.

From the obtained results, we observed that the calibration with respect to one selected measure improves not only this objective measure but also improves the remaining measures. See Tables 3–5, where we examined the Kharrufa, Hargreaves–Samani, and Turc model.

The measures presented in Section 2.5 are defined as the sum of local differences. Our results of cumulative evaporation presented in Figures 8–10 show the consequences of the formulation of the objective function in this form—the cumulative evaporation computed by the optimal calibrated model fits the cumulative evaporation computed by the *FAO* equation. We observed this property in the case of all measures. However, in the case of daily evaporation (or monthly evaporation), the local difference can be large, see Figure 11. Evaporation in some months has been underestimated and in other months has been overestimated. In any case, this underestimation and overestimation are always better than in the case of the original equation.

The obtained results follow the equivalency of the calibration process based on NSE maximization and RMSE minimization, i.e., Theorem 1. Please see Figures 5–7, where we demonstrate the density of optimal parameters of the calibrated Kharrufa, Hargreaves–Samani, and Turc equation with respect to the random data split in the cross-validation process. The small difference between NSE and RMSE is caused by the error of the iterative algorithm: the optimization algorithm has a stopping criterium based on the change of the function value. Since the NSE and RMSE have different objective functions, the iterative algorithm stops the optimization prematurely (sufficiently approximately) in different optimizers. Especially in the case of Figure 6, the difference is clearly observable. However, in this case, we are dealing with the Hargreaves–Samani model (19). We suppose that this difference is caused by the non-linearity of the model (and the non-linearity of used statistical measures). The difference between objective functions in the solutions computed by NSE and RMSE is approximately $10^{-2}$ (see Table 4), which is the value used in the stopping criteria of the iterative optimization algorithm. The situation is similar for RMSE.

The results obtained by our analysis show that the calibrated Hargreaves–Samani and Turc models seem to be the most suitable simplification of the FAO Penman–Monteith equation in the area of Lake Most. However, it is necessary to mention that the final choice of the most suitable calibrated equation for evaporation modelling depends not only on the final value of the statistical measures but also on the input data requirements. Therefore, we suggest using the Hargreaves–Samani equation since this equation requires only the input of the extraterrestrial radiation and the air temperature, see Equation (18). Figure 11 presents the final improved evaporation estimation.

## 5. Conclusions

In this paper, we presented the methodology for the calibration of evaporation models with the FAO Penman–Monteith equation and demonstrated it on selected simplified models using the most common statistical measures. Additionally, we implemented a cross-validation process to remove the overfitting of the calibrated model. This approach can be easily applied to any model of interest and any sufficiently reasonable statistical measure. In the paper, we presented a calibration with respect to theoretical values computed by FAO Penman–Monteith equation; however, the methodology can be used for calibration with any theoretical or measured reference values of evaporation.

From the presented results, we suggest using the Hargreaves–Samani equation to model the evaporation on Lake Most. This equation reported the sufficient approximation of the FAO Penman–Monteith equation and additionally, it requires only a few input parameters, which can be easily (and cheaply) measured.

During our research, we observed the global fitting property of common statistical measures—the evaporation during cold days is underestimated and the evaporation during sunny days is overestimated. To deal with this issue, we focus our future work on the division of days into groups with different optimal models.

**Author Contributions:** Conceptualization, V.D. and D.D.; methodology, V.D., L.P.; software, V.D.; validation, V.D.; formal analysis, L.P. and V.D.; investigation, V.D. and D.D.; resources, D.D. and V.D.; data curation, V.D. and D.D.; writing—original draft preparation, V.D. and D.D.; writing—review and editing, L.P.; visualization, V.D.; supervision, D.D.; project administration, D.D.; funding acquisition, D.D. All authors have read and agreed to the published version of the manuscript.

**Funding:** This paper has been completed thanks to the financial support provided to VSB-Technical University of Ostrava by the Czech Ministry of Education, Youth and Sports from the budget for conceptual development of science, research and innovations for the 2021 year and the Department of Mathematics at the Faculty of Civil Engineering, VSB-Technical University of Ostrava.

**Institutional Review Board Statement:** Not applicable.

**Informed Consent Statement:** Not applicable.

**Data Availability Statement:** The data are not publicly available due to the requirements of confidentiality and security. Data was obtained from The Institute of Atmospheric Physics CAS and are available from the authors with the permission of The Institute of Atmospheric Physics CAS.

**Acknowledgments:** The authors would like to express further thanks to The Institute of Atmospheric Physics CAS for the provided meteorological data from their measuring station Kopisty.

**Conflicts of Interest:** The authors declare no conflict of interest.

## Abbreviations

The following abbreviations are used in this manuscript:

| | |
|---|---|
| $E_{FAO}$ | FAO Penman–Monteith Equation (1), |
| $E_S$ | Šermer Equation (10), |
| $E_{BV}$ | Beran-Vizina Equation (11), |
| $E_{VUV}$ | the equation recommended by T. G. Masaryk Water Research Institute (12), |
| $E_K$ | Kharrufa Equation (16), |
| $E_{HS}$ | Hargreaves–Samani Equation (18), |
| $E_{Sch}$ | Schendel Equation (20), |
| $E_{PT}$ | Priestley–Taylor Equation (22), |
| PBIAS | Percentage Bias (29), |
| MAE | Mean Absolute Error (28), |
| RMSE | Root Mean Square Error (27), |
| NSE | Nash–Sutcliffe Efficiency (26), |
| $T_a$ | the average air temperature [°C], |
| $T_{max}, T_{min}$ | maximal and minimal air temperature [°C], |
| $T_r$ | the difference between daily maximum and minimum air temperatures [°C], |
| $P$ | atmospheric pressure [kPa], |
| $p$ | percentage of total daytime hours for the period used (daily or monthly) out of total daytime hours of the year. |
| $RH$ | relative humidity [%], |
| $R_a$ | the extraterrestrial radiation $\left[\text{MJ}\,\text{m}^{-2}\,\text{day}^{-1}\right]$, |
| $R_n$ | net radiation $[\text{kJ} \cdot \text{m}^{-2} \cdot \text{s}^{-1}]$, |
| $R_s$ | the solar radiation $\left[\text{MJ}\,\text{m}^{-2}\,\text{day}^{-1}\right]$, |

| $R_{so}$ | the clear-sky radiation $\left[\mathrm{MJ\,m^{-2}\,day^{-1}}\right]$, |
|---|---|
| $\gamma$ | the psychrometric constant $[\mathrm{m \cdot s^{-1}}]$, |
| $\sigma$ | Stefan-Boltzmann constant $\sigma = 4.903 \times 10^{-9}\,\mathrm{MJ\,K^{-4}\,m^{-2}\,day^{-1}}$ |
| $\lambda$ | latent heat of vaporization $\lambda = 2.45\,\mathrm{MJ\,kg^{-1}}$, |
| $C_p$ | the specific heat of air, $C_p = 1013\,\mathrm{J\,kg^{-1}\,{}^\circ C^{-1}}$, |
| $\rho_a$ | the air density $\left[\mathrm{kg\,m^{-3}}\right]$, |
| $\Delta$ | the slope of saturation vapor pressure curve $[\mathrm{kPa \cdot [{}^\circ C]^{-1}}]$, |
| $u_2$ | wind speed at height $2\,\mathrm{m}$ $\left[\mathrm{m\,s^{-1}}\right]$, |
| $G$ | the heat flow in the soil $[\mathrm{MJ\,m^{-2}\,day^{-1}}]$, |
| $e_s$ | mean saturation vapor pressure $[\mathrm{kPa}]$, |
| $e_a$ | the current water vapor pressure $[\mathrm{kPa}]$, |
| $e_s - e_a$ | vapor pressure deficit $[\mathrm{kPa}]$. |

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
