# Peer review of "Optimal Calibration of Evaporation Models against Penman–Monteith Equation"

_water, doi:10.3390/w13111484_

Round 1

Reviewer 1 Report

Dear authors,

Thanks for submission of this interesting and important article that has been prepared well.  I have only a few comments and suggestions below:

1) Some grammar and sentence structure including a few incomplete sentences were noted and need to be fixed. For example, L 33, L 94, L 129, L 221 ((14) should precede (15)), similar thing below eq (32) (should be stated as (28), (29), (30)), L 276-277, L 340-341; Table 3 caption middle statement: "..which respect to we are calibrating..." awkward  and should be fixed.

2) Statements mot clear and revise L  343-344; show this for Hargreaves-Samani also although "non-illustrative" to contrast for the readers?

3) L 279: Not quite clear where t(which Fig or Table) or paragraph, the results of Cross-validation methodology was presentation or discussed in Results Section.

4) In paragraph under eq 4, in statement with "....plant parts (such as stem, xylene (is this an appropriate term?, is it not xylem?, then use also "zero resistance" somewhere there to explain better! 

5) L 181: Why on what basis, "zm =10 m height is usual height of wind measurement?? Provide a reference because a standard height is 2 m.

6) L 129-130" what were the climate parameters measurement height at Kopisty weather station?

7) Several statements are suggested to be backed by some relevant references: L 19-20, , 44-45, L 115-122 for Lake Most , L 140-141, L 193-196, L 208-201.

8) Finally, I highly recommend the authors evaluate another widely TURC  PET method that uses temperature, humidity, and solar radiation.  That method should likely be applicable in that region and will complete almost all sorts of PET methods with different input variable combinations based on availability of data. A graphical Figure of mean monthly climate parameters they analyzed in their potential evaporation methods (similar to Figure 5 each plot for air temperature, humidity, wind speed, solar and net radiation) for their data period analyzed.

Thank you you.

Author Response

Dear reviewer, we would like to thank you for your interest in our work and we appreciate your comments. The paper is updated based on the comments from all reviewers. We did our best to address all your comments:

R1: Some grammar and sentence structure including a few incomplete sentences were noted and need to be fixed. For example, L 33, L 94, L 129, L 221 ((14) should precede (15)), similar thing below eq (32) (should be stated as (28), (29), (30)), L 276-277, L 340-341; Table 3 caption middle statement: "..which respect to we are calibrating..." awkward and should be fixed. A1: You are completely right. We did our best to fix sentences, please, see updated manuscript with highlighted changes. Additionally, we changed the order in presentation of the statistical measures to be the same as in tables. Now lists of equations used in the text are in order.

R2: Statements mot clear and revise L 343-344; show this for Hargreaves-Samani also although "non-illustrative" to contrast for the readers? A2: We extended presented results. The sentence is changed, now we discuss more results (both of illustrative and non-illustrative) to give reader an opportunity to decide what is illustrative and what is not.

R3: L 279: Not quite clear where t(which Fig or Table) or paragraph, the results of Cross-validation methodology was presentation or discussed in Results Section. A3: We extended the caption of figures by the information about used cross-validation. We hope that this information will give much more clear hint why the presented values are not just one number, but a random variable. We also extended Section "implementation", where we present more details of used cross-validation approach. We also included new illustrative figure.

R4: In paragraph under eq 4, in statement with "....plant parts (such as stem, xylene (is this an appropriate term?, is it not xylem?, then use also "zero resistance" somewhere there to explain better! A4: Xylene is a typo. We corrected this word in new version.

R5: L 181: Why on what basis, "zm =10 m height is usual height of wind measurement?? Provide a reference because a standard height is 2 m. A5: We changed word "usual" to "provided" (in Kopisty, the data are measured in this height). We added information about measurements to Section 2.2.

R6: L 129-130" what were the climate parameters measurement height at Kopisty weather station? A6: We added information about measurements to Section 2.2.

R7: Several statements are suggested to be backed by some relevant references: L 19-20, , 44-45, L 115-122 for Lake Most , L 140-141, L 193-196, L 208-201. A7: We added references. The problem is that the most of the available information are in Czech language. We added a reference to the webpage of PKU, which is securing the area of Lake Most.

R8: Finally, I highly recommend the authors evaluate another widely TURC PET method that uses temperature, humidity, and solar radiation. That method should likely be applicable in that region and will complete almost all sorts of PET methods with different input variable combinations based on availability of data. A graphical Figure of mean monthly climate parameters they analyzed in their potential evaporation methods (similar to Figure 5 each plot for air temperature, humidity, wind speed, solar and net radiation) for their data period analyzed. A8: We would like to thank you for a suggestion. We included TURC model to our comparison. From our results, we can see that this model is truly appropriate for our interests. However, there is larger number of input data, which has to be also considered. Please, see updated manuscript with updated figures for new results and discussion. In the new version of the manuscript, we highlighted all the changes using the red color.

We are not native English speakers. For the purpose of English language and style improvement, we would like to make use of the MDPI editing services.

With best regards, authors

Reviewer 2 Report

The paper concerns the different methods pf calculating eveporation, which are not so data demanding as Penman-Monteith equation. The methodology of selection of best method in particular region is proposed. It is important issue, however the methodology of calibration and implementation cross-validation process to remove the overfitting of the calibrated model is very poorly described and it is impossibly to follow the authors’ activities in this section.

Author Response

Dear reviewer, we would like to thank you for your interest in our work and we appreciate your comments. The paper is updated based on the comments from all reviewers. We did our best to address all your comments:

R1: the methodology of calibration and implementation cross-validation process to remove the overfitting of the calibrated model is very poorly described and it is impossibly to follow the authors’ activities in this section. A1: We tried to do our best to describe the calibration process in new version of the manuscript. Please, see the text in Section 2.6., where we highlighted all the changes using the red color. Additionally, we included the new figure illustrating the process.

We are not native English speakers. For the purpose of English language and style improvement, we would like to make use of the MDPI editing services.

With best regards, authors

Reviewer 3 Report

First off, thanks to the authors for the care of this submission.  I thought the writing, organization, figures and tables were very refined and I'm very grateful as a reviewer.

* Biggest concern

I am definitely confused about what PM equation you are using for comparison.

EQ 1. Indicates the FAO ETo, (for reference surface grass), but then you
start w/ EQ 2, and show estimates, (eg. r_s=0 r_a, Rn) that are for open water, and not for reference surface. So the formulations in L140-L201, do not lead to Eq 1. (again r_s,r_a,Rn) (depending on what albedo you mean for Rn)

Again in L179, you say reference grass type crop, but use open water for r_s , r_a and albedo in Rn.

So big question is PM base on EQ1, with all reference surface( r_s,r_a,Rn) or EQ2 with the open water values for above. I would assume you'd use the EQ2, for open water, because then these optimizations that you are doing below are specific to open water evaporation, and not just a comparison of ETo that happens to be by a lake.

L183, if evaporation depends on lake depth (T_water?, Rn) then either PM formulation needs adjustment.
In L183-192 you discuss K factors for ETo (eq1) w.r.t to water, but if you are using PM_openwater (EQ2), then maybe you can more reasonably say K=1 and your equations can lead to evaporation estimates.

= Minor Issues ==

Is it possible to calculate direct evaporation measurement vs. PM from data used in Eq 13-16? or do they just have Ta?

Table 2. I thought RMSE and NSE are supposed to have same optimization? Why are their parameters different on some of the models? Is that what you say in L 332? Maybe add as a note to the table?

Paragraph @ L346, not really important.

In your conclusions, you say that optimized HS is best for Lake Most.
L89: This is for other lakes, (Most lake has 308 ha^2, avg depth 75m)

This is accurate#e, but in L89. You say the point is to apply to other lakes, and in previous work, you note other calibrations based on lake Temp, and/or depth. So when can you apply this optimized version in place of the un-optimized HS equation?

Also, since it's a closed system, do you not have *ANY* direct evaporation measurements to compare the estimates to?

HS ET is best solution, least amount of change via optimization. That's an important conclusion as well IMO.

Fig 1: Right hand picture should be closer on lake, would be great to have a scale on it as well. Maybe show: meteorological station Hlasivo as well?

Fig 5. I can't read the top graph very well. The lower one alone (Or just one specific year) would be fine IMO.

Author Response

Dear reviewer, we would like to thank you for your interest in our work and we appreciate your comments. The paper is updated based on the comments from all reviewers. We did our best to address all your comments:

R1: I am definitely confused about what PM equation you are using for comparison. EQ 1. Indicates the FAO ETo, (for reference surface grass), but then you start w/ EQ 2, and show estimates, (eg. r_s=0 r_a, Rn) that are for open water, and not for reference surface. So the formulations in L140-L201, do not lead to Eq 1. (again r_s,r_a,Rn) (depending on what albedo you mean for Rn) Again in L179, you say reference grass type crop, but use open water for r_s , r_a and albedo in Rn. So big question is PM base on EQ1, with all reference surface( r_s,r_a,Rn) or EQ2 with the open water values for above. I would assume you'd use the EQ2, for open water, because then these optimizations that you are doing below are specific to open water evaporation, and not just a comparison of ETo that happens to be by a lake. A1: We are using Equation 1 as an etalon for the evaporation estimation, we calibrate all models against this equation. Equation 2 is used as a source from which Equation 1 has been derived. We modified the text and added a new discussion. Please see the end of the Section 2.3. In the manuscript, we distinguish between Penman-Monteith equation (Eq 2) and FAO Penman-Monteith equation (Eq 1). We checked the whole text for right references.

R2: L183, if evaporation depends on lake depth (T_water?, Rn) then either PM formulation needs adjustment. In L183-192 you discuss K factors for ETo (eq1) w.r.t to water, but if you are using PM_openwater (EQ2), then maybe you can more reasonably say K=1 and your equations can lead to evaporation estimates. A2: Similarly to the previous answer, please, see new discussion in the end of Section 2.3.

R3: Is it possible to calculate direct evaporation measurement vs. PM from data used in Eq 13-16? or do they just have Ta? A3: Yes, these equations has been derived to compute evaporation estimation. However, these equation has been originally derived to compute the estimation for different locations. In the paper, we are calibrating the parameters to obtain better estimation (in comparison to FAO) in the area of Lake Most. Thank you for your notice, we removed empty equation (16).

R4: Table 2. I thought RMSE and NSE are supposed to have same optimization? Why are their parameters different on some of the models? Is that what you say in L 332? Maybe add as a note to the table? A4: Theoretically, RMSE and NSE calibration optimization problems have the same solution, which we proved in Theorem 1. In the proof, we do not suppose anything special about the objective function, except the existence of the solution. The proof is based only on the zero-order optimality condition (i.e., the function values in all feasible points is less or equal than the value in the optimizer). However, we are solving the problem numerically and therefore the approximation of the solutions can be different, especially when the objective function is highly non-linear. We included new tables with final optimized statistical measures for Kharrufa, HS, and Turc, see Tables 3,4,5 in the new version of the manuscript. We also included new results, which show that (in the case of using numerical algorithm) the optimizers are different. Please, see Figure 6 (in new manuscript), where we present results for Hargreaves-Samani model. The difference is further discuss in section Discussion.

R5: Paragraph @ L346, not really important. A5: The paragraph has been removed. You are completely right - the small amount of used data does not require any special comment about parallel implementation.

R6: In your conclusions, you say that optimized HS is best for Lake Most. L89: This is for other lakes, (Most lake has 308 ha^2, avg depth 75m) This is accurate, but in L89. You say the point is to apply to other lakes, and in previous work, you note other calibrations based on lake Temp, and/or depth. So when can you apply this optimized version in place of the un-optimized HS equation? A6: Please, see line 224 in new manuscript, where we are discussing the extension of our research to other lakes in the area. Based on the conclusions of the paper, we are suggesting to use calibrated HS also to other planned lakes because it requires only the input of temperature measurements. In this case, these lakes are even more distant from Kopisty meteostation. In these case, we suppose that the measurements obtained in this meteostation will be not sufficient to provide correct evaporation estimations without measuring some input data directly in the closer area. The measurement of the temperature is the cheapest option. Please, see the extension of the discussion provided in the new version of manuscript.

R7: Also, since it's a closed system, do you not have *ANY* direct evaporation measurements to compare the estimates to? A7: To be honest, we have "some" data. However, we have not been able to process them into the paper. Some of the data are missing (the evaporation has been measured in Lake Most from 13.7.2017 to the end of 2019 = approximately 900 days, out of this it was 120 day out of order), some of them are corrupted by huge measurement errors. We are doing our best to obtain at least some information from those data and we hope that we will have an opportunity to present our findings in another publication.

R8: HS ET is best solution, least amount of change via optimization. That's an important conclusion as well IMO. A8: We extended our results and the discussion. Additionally, we included Turc equation to our analysis (has been suggested by one reviewer). This equation looks also promising, however, it requires more input data.

R9: Fig 1: Right hand picture should be closer on lake, would be great to have a scale on it as well. Maybe show: meteorological station Hlasivo as well? A9: We changed the map. We are using the meteorological data from Kopisty station. The data from Hlasivo station has been used by authors of models presented in Section 2.4.1. Unfortunately, Hlasivo station is situated around 140km from Lake Most. Probably this distance is the reason why the original models derived on the data from this location does not fit on Lake Most.

R10:Fig 5. I can't read the top graph very well. The lower one alone (Or just one specific year) would be fine IMO. A10: In this Figure, we would like to demonstrate the periodical behaviour of the evaporation. The bottom Figure does not present the evolution of the evaporation during the year. In the new version of the manuscript, we highlighted all the changes using the red color.

We are not native English speakers. For the purpose of English language and style improvement, we would like to make use of the MDPI editing services.

With best regards, authors

Round 2

Reviewer 1 Report

Dear authors,

Thank you very much for addressing almost all of my comments in this revision. I am glad that you also added Turc method and was found to perform better than other models. I understand your conclusion based on your objective to chose the one with the least parameters (H-S in this case) as well as the evaluation statistics for the model performance.

I still found few minor grammatical errors that I suggest to be easily fixed in the revised areas (in Red font) I noticed as follows:

L 147-148" Add "above the ground" after "10 m" also should be "at 10 m above …" and "at 2 m above..." not "in 10 m " or "in  2m:.  Replace "in" with "at"

L 50-51: both references are somewhat old although good ones. Suggest that you also add  recent 2016 one by

     Jensen ME, Allen RG (Eds). 2016. Evaporation, Evapotranspiration, and Irrigation Water Requirements. 2nd Edition, Manuals and Reports on Engineering Practice No. 70, New York, (NY): American Society of Civil Engineers

L 177: Replace "in" with "at" here also.

L 225: Replace "focus" with "focused"

L226: Replace "not exist in the presence" with "does not exist at present"

Really suggest to be thoroughly checked one more time throughout the manuscript.

Other than this I do not have any more comments.  Thank you all again.

Author Response

Dear reviewer,

we would like to thank you for your positive feedback and new comments. 

R1: L 147-148: Add "above the ground" after "10 m" also should be "at 10 m above …" and "at 2 m above..." not "in 10 m " or "in  2m:.  Replace "in" with "at"
A1: corrected

R2: L 50-51: both references are somewhat old although good ones. Suggest that you also add  recent 2016 one by
    Jensen ME, Allen RG (Eds). 2016. Evaporation, Evapotranspiration, and Irrigation Water Requirements. 2nd Edition, Manuals and Reports on Engineering Practice No. 70, New York, (NY): American Society of Civil Engineers
A2: reference added

R3: L 177: Replace "in" with "at" here also. L 225: Replace "focus" with "focused". L226: Replace "not exist in the presence" with "does not exist at present".
A3: corrected

R4: Really suggest to be thoroughly checked one more time throughout the manuscript.
A4: In the second round of reviews, we focused on English and spellcheck. We checked the whole text using the spell checker. We hope that this process will eliminate misprints and improve grammar. We are not native English speakers, but (at least) we are trying to do our best.

In the new version of the manuscript, we removed the red colour from the first round of reviews and highlighted all the changes using the red colour in this round.

With best regards,

authors 

Reviewer 2 Report

I am satisfied with corrections provided by authors

Author Response

Dear reviewer,

we would like to thank you for your positive feedback. 

In the second round of reviews, we focused on English and spellcheck. We checked the whole text using the spell checker. We hope that this process will eliminate misprints and improve grammar. We are not native English speakers, but (at least) we are trying to do our best.
In the new version of the manuscript, we removed the red colour from the first round of reviews and highlighted all the changes using the red colour in this round.

With best regards,

authors 

Reviewer 3 Report

Thanks again to the careful consideration of the authors on this article.

Author Response

(The authors gave the same response as above.)
